# Contemporary Landscape Structure within Monumental Zone-1 at Bagan Cultural Heritage Site, Myanmar

**Min Zar Ni Aung** *,† and **Shozo Shibata** †

Laboratory of Landscape Architecture, Division of Forest and Biomaterials Science, Graduate School of Agriculture, Kyoto University, Kyoto 606-8502, Japan; min.aung.28r@st.kyoto-u.ac.jp
* Correspondence: minzarniaung@gmail.com
† These authors contributed equally to this work.

**Abstract:** This study examines the contemporary landscape structure of the Monumental Zone (MZ)-1 at the Bagan Cultural Heritage Site in the Dry Zone of Myanmar. With respect to hundreds of medieval monuments, how local residents in the residential areas within the MZ-1 manage the landscape was the focus of the current study, conducted with two objectives: (1) Identifying land covers as features of the contemporary landscape on the basis of land use and (2) evaluating how the features interrelate. The landscape features were identified by the analysis of Landsat 8 satellite imagery, followed by variance analysis for comparison of the features' areas, and interrelationships of features were evaluated by multivariate analysis. Vegetated features were identified in coexistence with non-vegetated ones, while crop coverage and non-vegetated features were smaller than the area of the other two vegetated features. Semi-natural woody vegetation was found in proximity to monuments and was dependent on the occurrence of the shrub-prone patch that, in turn, was triggered by the expansion of exposed land containing a large segment of cultivatable area. The current study suggests the need to prioritise timely land use and management, focusing on local agricultural activity for safeguarding the heritage as well as the historical settings.

**Keywords:** Bagan; Myanmar; monument; cultural heritage; landscape features and structure; unmanaged vegetated cover; anthropogenic artefacts; agricultural land

---

## 1. Introduction

Landscape, at a given spatial scale, fundamentally represents a heterogeneous composition of biophysical, social, and cultural objects developed by natural means and/or human interventions on the earth's surface [1,2]. These objects in a landscape scale can be seen as various landforms, vegetation patches, water bodies, settlement areas, roads, etc., and are referred to as land covers or features [3,4]. The features in spatial configuration make up the landscape structure [1,5], giving an identity to the landscape that can possess a distinctive appearance, in contrast to the surroundings [6]. Nevertheless, the natural pressure, including storm, flood, fire, earthquake, and/or anthropogenic land use and management, are all drivers that modify the landscape and make it unstable [6,7].

Modification of the landscape by humans has been rather significant across the globe, and, in consequence, it yields an impact on the landscape structure as well as on the interaction between landscape features [8]. However, research on the ecological aspects of landscape is very limited in tropical Asia [9]. The current study is a preliminary effort to develop deeper knowledge of ecosystem services and management through safeguarding the cultural heritage at the Dry Zone of Myanmar.

The Bagan Cultural Heritage Site (BCHS) is home to 3822 medieval-aged Buddhist monuments—stupas (solid-type pyramidal structures), temples, and monasteries [10–13]. The entire

stretch of the BCHS is 22,809.85 ha, composed of eight monumental zones (MZ), together with other components, and it expands over the eastern and western banks of the Ayeyarwady River [14]. The MZ is defined by the presence of monuments within a designated area and, of the eight zones, the MZ-1 accounts for more than 2000 monuments [15–17].

The presence of more than 50% of the region's total monuments makes the MZ-1 a unique landscape. The monuments were built during the 11–13th centuries [18] by early settlers in the Bagan area, and such activities are regarded as attempts to transform the terrain into a cultural landscape [19]. In the present era, there are seven residential areas in the MZ-1, and agricultural lands managed by the residents in the environs of the monuments endow the zone with a living landscape. This study's focus, therefore, centres on how residents in those areas manage the landscape with respect to the hundreds of monuments as safeguarding measures for tangible and intangible cultural heritage. Agricultural land management, in explicit postulation, is a trigger of impacts on the environmental setting of the heritage site; therefore, management-related data to date are a prerequisite for long-term analytical studies.

Within the Dry Zone of Myanmar, people's livelihoods are primarily dependent on agriculture, with 57% of the land altered through cultivation [20]. Nevertheless, global climate change has also affected agricultural production for people in the dry zone, due to irregular rainfall [21,22]. Moreover, experiencing life in the Dry Zone, the MZ-1 settlers have experienced the ripple effects of the adverse climate on their agricultural activities. If crop production declines, the land used for cultivation by local farmers is likely to be minimized. Reducing cultivated land implies shrinking crop cover in one word, and expanding weedy cover and/ or shrub on the agricultural land in other words. The current conditions of agricultural land use might have created some distinctive characteristics regarding landscape features at the MZ-1. The speculation, therefore, was a matter of the analytical study on contemporary landscape structure, based on two specific objectives: (1) To identify landscape features as land cover on the basis of land use, and (2) to evaluate the interrelationship of features at the MZ-1 in terms of spatial coverage.

The status of features, as well as land use impacts, was first assessed by satellite image analysis, and, second, the interdependence among the features was evaluated by multivariate analysis. The study shed light on vegetated and non-vegetated features as a primary structure of the landscape of the MZ-1, reflecting four major components in coexistence—"unmanaged vegetated cover", "anthropogenic artefacts", "agricultural land", and "reservoir". The results of the current study stress the important role of site management, explicitly, agricultural land, due to massive expansion of unmanaged vegetated cover in landscape scale. Regarding well-rooted conservation and preservation measures for the monuments and the site, the study suggests the vegetation in the zone to be under management for strengthening integrity of the cultural landscape.

## 2. Materials and Methods

### 2.1. Study Site

The study site for this research was the MZ-1 in the BCHS, which extends 21.10093–21.19742 N, 94.84953–94.91978 E in the central Dry Zone of Myanmar (Figure 1). Among the eight MZs of the BCHS, the MZ-1 stands apart as the largest repository of more than half of the total monuments of the BCHS, on its expansion to 4164 ha. The MZ-1 occupies 18% of the land surface in the BCHS, while the other 7 MZs, 2 urban areas (Nyaung-U and New Bagan or Myo-thit), 37 villages, and the buffer zones account for 82%. The topography of the MZ-1 characterizes massive stretches of level terrain, despite the presence of a few depressions for reservoirs.

At the study site, six villages are composed of four village-tracts, comprising "Bwasaw", "Min Nan Thu", "Taungbi-Leya" and "Myinkaba" in Nyaung-U Town, and one ward in New Bagan Town. According to the National Population and Housing Census of Myanmar (2014) [23], the total number of residents of these areas is 15,706, contributing to 8% of the Nyaung-U Township population. The residents' primary source of income is through farming, whereas crafts, tourism-related businesses,

and seasonal labour supplement their earnings. Being situated in the drier part of Myanmar, where the scant annual rainfall rarely exceeds 1000 mm [24], the land in the MZ-1 primarily favours production of oil-seed crops and pulses. The farmlands for these crops spread mainly in the environs of monuments, coexisting with semi-arid vegetation.

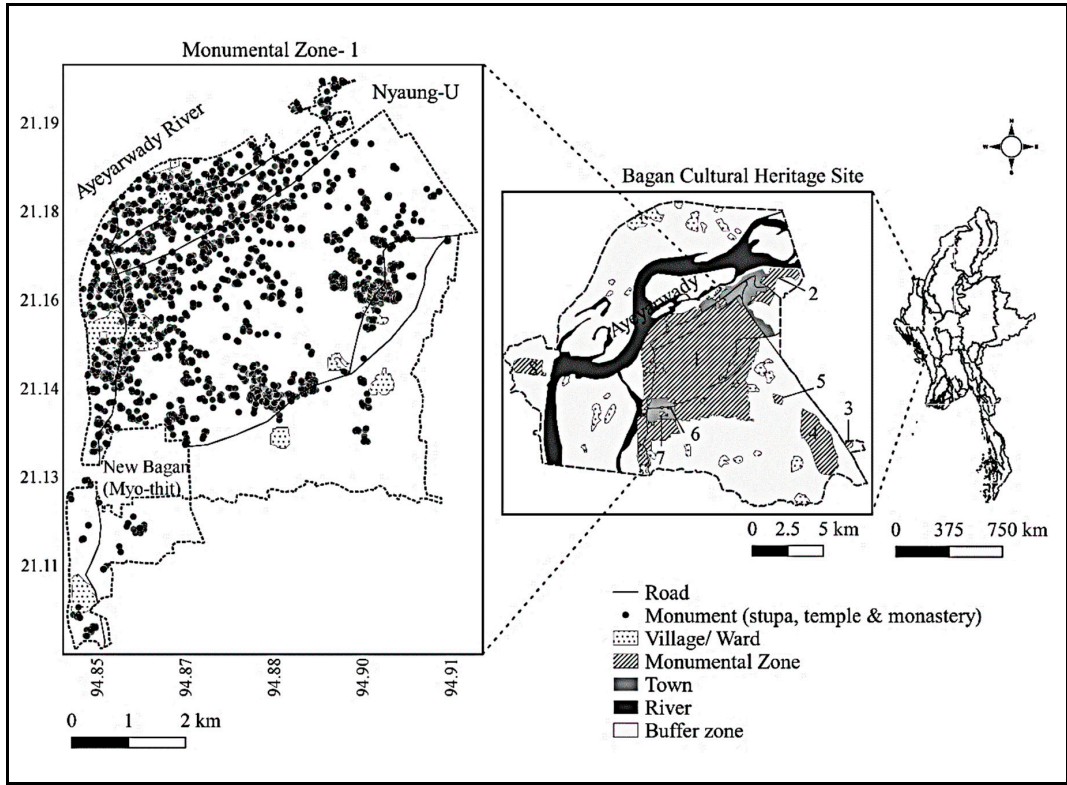

**Figure 1.** Location of Monumental Zone (MZ)-1, Bagan Cultural Heritage Site (BCHS) at central Dry Zone of Myanmar. Source [14,25].

## 2.2. Field Survey

Two field surveys were performed for spatial knowledge of the study site: The first in August–September of the rainy season in 2017 and again in 2018. Across the MZ-1, geolocations of zone boundary, monuments, fields covered with crops and shrubs, ploughed and/or exposed lands, patches of semi-natural woody vegetation (woods that have not been planted, but occur due to human interference), roads, settlements, and reservoirs were recorded with a handheld Global Positioning System (GPS) (Oregon®700, Garmin). The locations of the monuments were crosschecked with the inventory list developed by Pichard (1992–1995) [12]. During the field surveys, monuments in inaccessible locations were assessed by satellite image (2 m resolution) from Google Earth (CNES/ Airbus, 2018), and points for 2348 monuments and unexcavated mounds were recorded within the MZ-1 (Figure 1).

## 2.3. Image Processing

Multiband satellite images from Landsat 8 (Optimal Land Image (OLI), 8 Bands, 30 m resolution) were acquired for 28 September, 2018, in cloud free conditions, from the United States Geological Survey (USGS) [26]. Although September is late in the rainy season within the Dry Zone of Myanmar, farmers generally maintain a cover for the cultivatable land with crops and vegetation, and cover can also be of good foliage mass. The image (Landsat scene ID: LC81340452018271LGN00) covered the overall extent (upper left corner, 22.72132 N, 93.61154 E; upper right corner, 22.69764 N, 95.81883 E; lower left corner, 20.62895 N, 93.60277 E; lower right corner, 20.60766 N, 95.77850 E) of the site integrated into the study.

The images were pre-processed by raster clipping into the MZ-1, followed by generating reflectance images using Bands 2, 3, 4, 5, 6, and 7 in a semi-automatic classification plugin (SCP, version 6.2.7 Greenbelt) of the Quantum Geographic Information System (QGIS 3.4.2). Pan-sharpening of the image, using Bands 2, 3, 4 and 8, was also processed for reference images in accuracy checks after land cover classifications. A pan-chromatic image of Landsat 8, 15 m resolution (Band 8), was blended with the other three bands, and the pan-sharpened image generated was of high resolution (15 m).

Bands of the reflectance images were computed for the Modified Soil Adjusted Vegetation Index (MSAVI) as follows: MSAVI = {(NIR − RED)/ (NIR + RED + L)} × (1 + L), where NIR and RED of the images are near infrared (Band 5) and red (Band 4), respectively [27]. The "L" represents the soil adjustment factor equal to 0.5, fit for the broad range of vegetation cover, and was used in computation to suppress interference of background soil reflectance, due to sparse vegetation density at the study site [28]. The MSAVI, therefore, was applied to improve the surface reflectance of vegetation cover where it existed across the MZ-1, while reducing soil influence on the index value.

For the presence of residential areas, monuments, and paved roads as built-up features at the study site, image interpretation with MSAVI alone was doubtful in displaying all land covers. In this regard, the image was further analysed with the Normalized Difference Built-up Index (NDBI): NDBI = {(SWIR − NIR) / (SWIR + NIR)} [29], where SWIR is short-wave infrared-1 (Band 6). The vegetation index (MSAVI) was subtracted from the NDBI to delineate the built-up features that suppress the vegetation influence in this study. As described by Zha et al. (2003) [29], the ploughed and/or exposed land, which is part of the agricultural land and will be termed "exposed land" henceforth, displayed positive value, similar to that of the built-up features. In addition, water features were interpreted with a Normalized Difference Water Index (NDWI): NDWI = (Green − SWIR) / (Green + SWIR) [30], where green is Band 3. The image assessed with NDWI enhanced water surfaces in order to differentiate them from other land covers. When compared with the image of MSAVI and checked against ground-truth records, the features interpreted with NDBI and NDWI were found to be the same as those displayed with MSAVI.

Following the image interpretation with the indices, the land covers and features were classified, using the supervised classification by the SCP, according to Congedo (2016) [31]. The algorithm generated a representation of the maximum likelihood of the total features by segmenting them into classes via computing pixel similarity [32]. Once classified, the features were checked for accuracy by means of an error (fusion) matrix with randomly selected polygons (pixels) for respective features from the pan-sharpened reference image (15 m resolution). The polygons for the feature ranged from 20–120, and, before accuracy assessment, they were checked against field-recorded GPS points in a 2-m resolution Google Earth map (CNES/Airbus, 2018). The smaller number of polygons than those representing the other features were water and cultivated field, the former of which will be termed "reservoir" and the latter considered "crop field", hereafter. Land use for each feature was further analysed, following an accuracy check.

In order to evaluate the interrelationships of each feature with the others and as a crosscheck for the accuracy of classification, pixels of respective features were counted manually in a 30 m resolution Landsat 8 multiband image. The pixels were marked with regular points of 50 m distance so that two adjacent pixels accounted for corresponding features. For precision of counting, grids were applied to the image, and the size of each grid cell was 1 km$^2$. Within each grid, 400 points marked the pixels, and 8562 pixel counts of the six features were accessed in 30 grid cells that completely and maximally overlaid the image. Some grids at the boundary of the image did not encompass the marked points completely, thus, the points in these parts gave no account. In this case, locations of monuments digitized as point data were also added to the image, and many points of the monuments fitted to the pixels were counted. As an estimate of area for each feature, the extracted pixel was multiplied by its area (900 m$^2$) [33]; the data were then applied to verify the tendency of land use for each feature generated by supervised classification.

*2.4. Data Analysis*

The calculated pixel areas of the six features were assessed for analysis of variance (ANOVA) with Duncan's Multiple Range Test (DMRT) at a significant level ($\alpha = 0.05$). The point-data of the features counted, including the monument, were processed for multivariate analysis (principal component analysis {PCA}). The statistical analyses were implemented by SAS University Edition.

## 3. Results

*3.1. Landscape Features and Land Use*

The contemporary landscape of the MZ-1 was interpreted by five index ranges of MSAVI (Figure 2) into two major types of land cover or feature: Vegetated and non-vegetated. The vegetated covers depicted the index values in descending order, from 0.662 to 0.272, and represented crop fields, semi-natural woody vegetation, and shrub-prone patches. The non-vegetated covers ranged from 0.246 to −0.223 and included built-up area, exposed land surface, and water.

The crop fields were found under coverage of oil-seed crops, pulses, and sorghum, which reflected high indices, between 0.558 and 0.662. The semi-natural woody vegetation consisted predominantly of xeromorphic thorny plants, and it indexed between 0.350 and 0.532. Nevertheless, a patch of broadleaf woody plants in the "Lawka Nanda" Wildlife Sanctuary responded with similar indices to the crop field. Shrub-prone patches characterized the lands where crop cultivation was practised in previous years, but they still represent part of agricultural land. This feature ranged between 0.272 and 0.324 among the indices of vegetated covers.

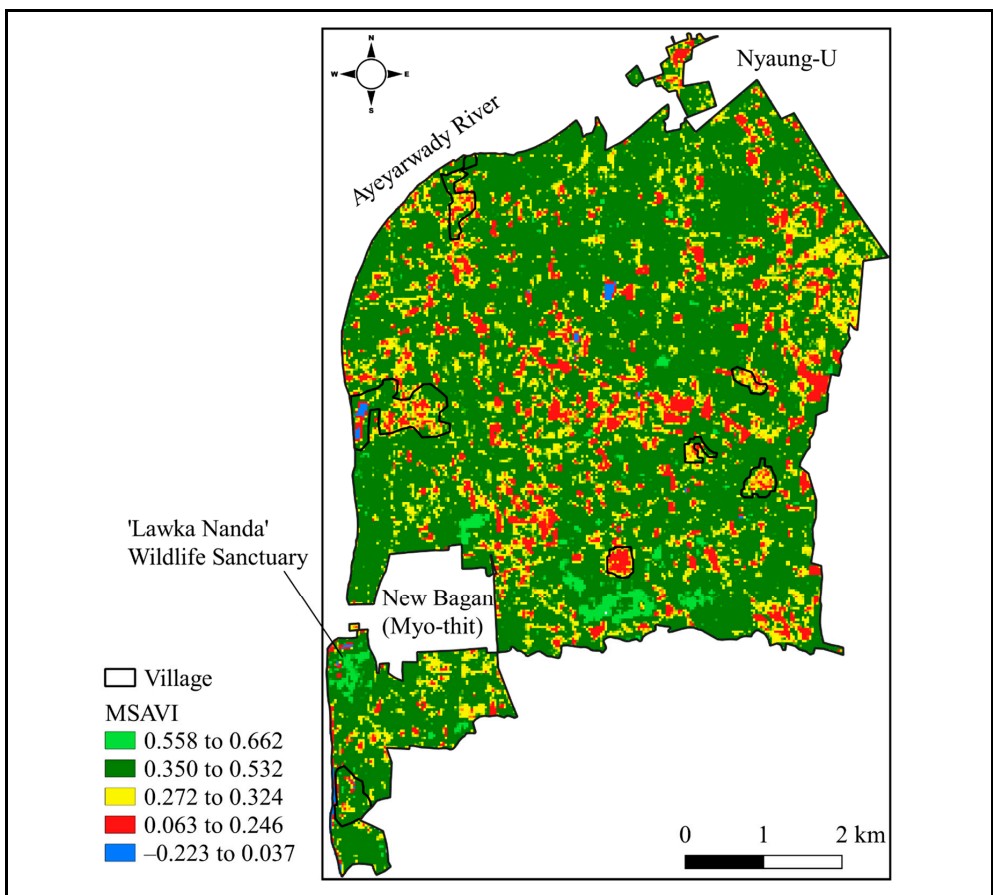

**Figure 2.** Landscape features (land covers) of the Monumental Zone-1 in the Bagan Cultural Heritage Site. Features interpreted with the Modified Soil Adjusted Vegetation Index (MSAVI) by five index ranges. The index range, 0.350 to 0.662, represents vegetated cover, and −0.223 to 0.324 reflects non-vegetated cover. Source [26].

The interpretations of landscape features did present some ambiguities. In fact, the buildings in the villages resembled exposed land surfaces, possibly for their closeness of indices. Irrespective of the village area, these two features are supposed to be homogeneous (Figure 2); therefore, the built-up features were further assessed with NDBI (Figure 3). However, the built-up area was indistinguishable from the exposed lands in the range of NDBI, 0.322 to 0.234, while other features were suppressed. Paved roads as built-up structures were not supported by the NDBI, and they were seen instead under the influence of semi-natural woody and shrub-prone features (Figure 2). Similarly, monuments were not sharply interpreted by the NBDI in the study. Thus, cautious interpretation here was applied, based on local knowledge in order to discriminate the features correctly. The water reflected from near zero to negative MSAVI (0.037 to −0.223); therefore the existence of water was affirmed in the zone with NDWI, ranging from 0.334 to 0.943 in which other features were not recognized (Figure 4).

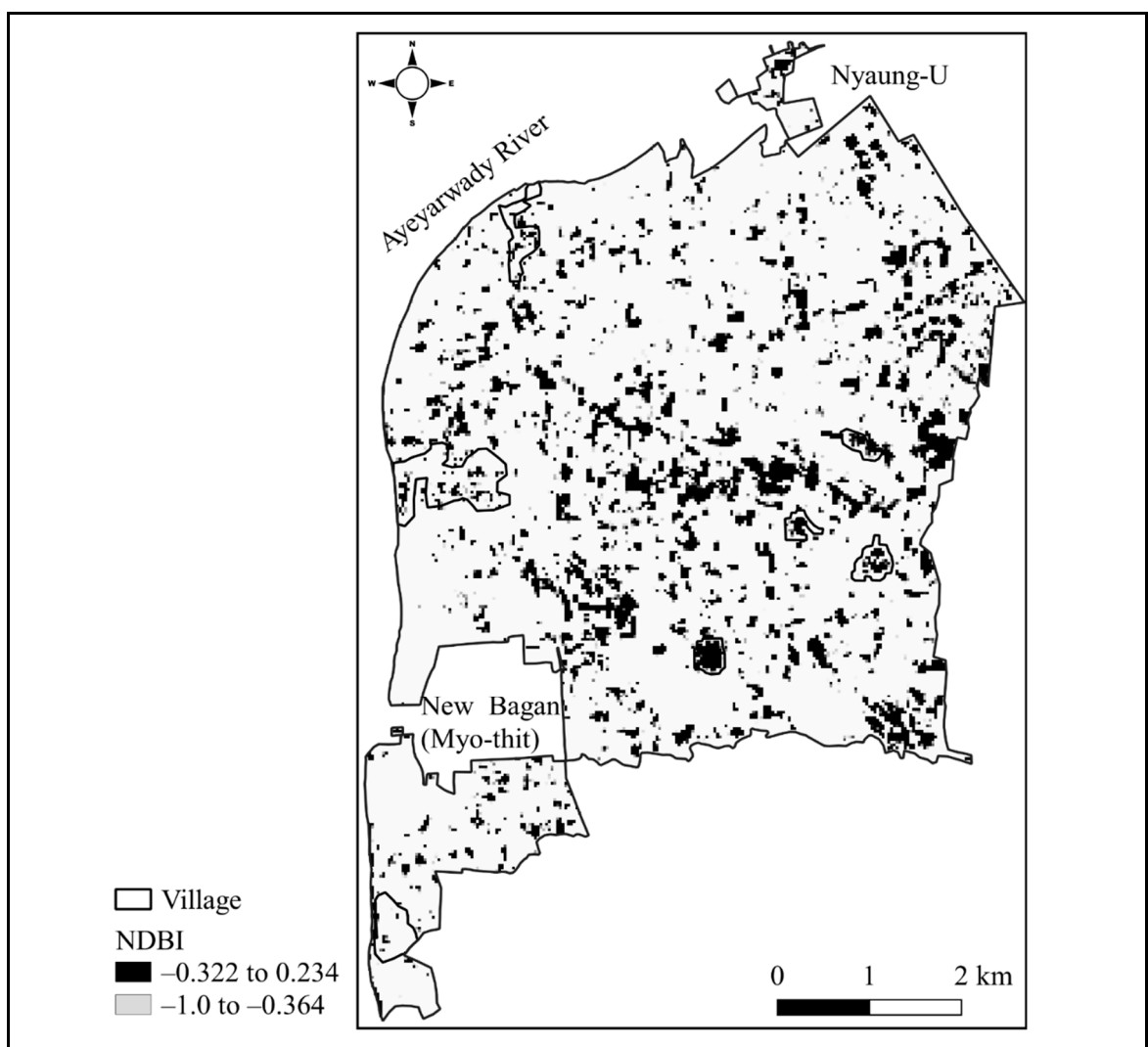

**Figure 3.** Buildings in the villages of the Monumental Zone-1 interpreted with Normalized Difference Built-up Index (NDBI) (−0.322 to 0.234), in contrast with non-built-up features (−1.0 to −0.364). Residences in the villages resemble exposed lands across the zone, and some are influenced by vegetated features. Source [26].

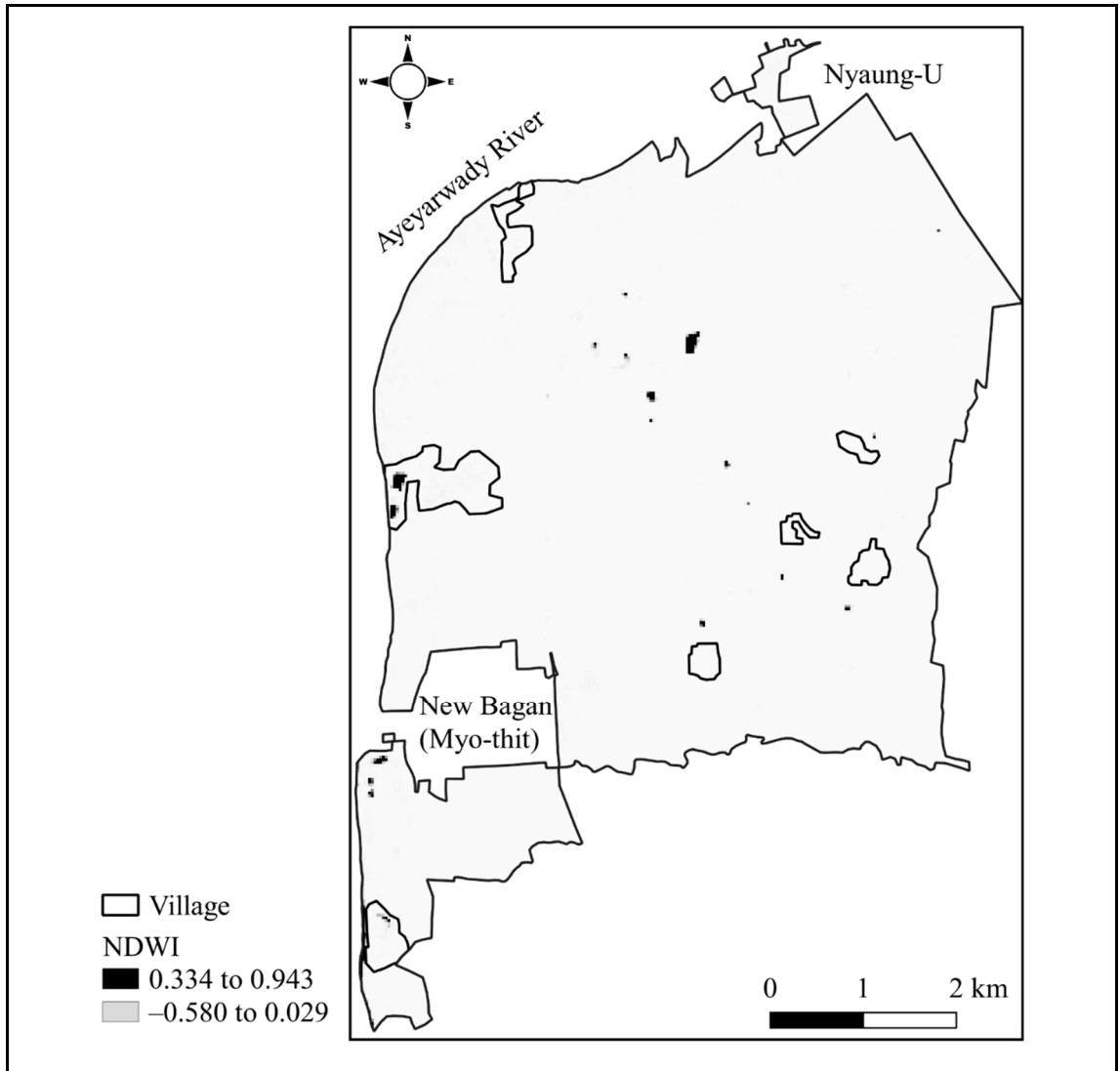

**Figure 4.** Water surfaces at the Monumental Zone-1 interpreted with Normalized Difference Water Index (NDWI) (0.334–0.943), contrasting non-water features of NDWI 0.029 and −0.580. Source [26].

The index-based interpretation was followed by supervised classification with SCP to evaluate landscape features and land use. As a result, six features were classified as land cover, with an overall accuracy of 82% (Kappa index 0.8), and, in addition, built-up features were distinguished from exposed land in another four (Table 1). The features were as accurate as 71.6–100% in terms of producer accuracy and 69.4–100% according to user accuracy. The Kappa indices for the agreement between producer and user ranged from 0.64 to 1, indicating that the features were classified with high accuracy. Of the six land covers, only the built-up features were around 70% accuracy in each case, with a Kappa index of 0.64, while the other five were accurate by above 70%.

The land use for the crop field was found to account for 7.5% in terms of relative area, whereas those for the semi-natural woody vegetation and the shrub-prone patches were 37.1% and 25.8%, respectively (see Table 1). These three features representing vegetated cover occupied 2933.91 ha, or 70.5% of the MZ-1. In other words, proportions of the built-up features, the exposed land, and the water were 13.5%, 14.7%, and 0.4%, respectively, which make up 28.6% of the non-vegetated area, or 1189.62 ha. The data in such circumstances showed that a small proportion of land was used for cultivated crops, while there were about 2–5 times larger land parcels covered by the other features, exempting water. The overall area covered by the six features accounted for 4123.53 ha,

equivalent to 99.1% of the MZ-1; otherwise, the discrepancy of less than 1% was unidentified by supervised classification.

The ANOVA of mean pixel areas calculated from pixel counts for respective features indicated that the coverage of the crop field was, compared with those of the semi-natural woody vegetation and the shrub-prone patches, 6-fold and 3-fold smaller, respectively (Table 1). The mean area of the crop fields, however, was not different to that of the built-up features, the exposed land, or the water, while the built-up area and the exposed land occupied an area about an order of magnitude wider than the water. The results suggested that the current land cover by the crop field appeared as a comparatively small area, whereas other vegetated covers were massive. Having a similar statistical relationship to the crop field, the built-up area and the exposed or ploughed land were as large as the areas covered with crops. The land used for crops seems to be a small fraction of the total, as does the water surface, whereas the exposed land was part of the cultivatable area, though it was not covered with crops. In other words, the contrast in mean area among the features is arguable in that the land use for one feature relates with others that coexisted at the study site.

**Table 1.** Landscape features (land covers) at the Monumental Zone-1.

| Feature | Supervised Classification [†] | | | | | Estimated Area [††] by Pixels | | |
|---------|-------------------|-----|-----|-----------|----------------------|-------------|----------------------|-------------|
| | Accuracy | | | Area (ha) | Relative area (%) | Mean (ha) | Relative area (%) | Pixel count |
| | Producer (%) | User (%) | Kappa Index | | | | | |
| Crop field | 97.39 | 77.00 | 0.74 | 313.38 | 7.53 | 2.13 [cd] (± 0.61) | 7.75 | 450 |
| Semi-natural woody vegetation | 92.29 | 86.09 | 0.81 | 1545.30 | 37.11 | 13.04 [a] (± 1.14) | 47.44 | 4348 |
| Shrub-prone patch | 77.11 | 88.95 | 0.85 | 1075.23 | 25.82 | 5.93 [b] (± 0.65) | 21.57 | 1976 |
| Built-up feature | 71.56 | 69.36 | 0.64 | 560.97 | 13.47 | 2.93 [c] (± 0.67) | 10.66 | 813 |
| Exposed land | 73.67 | 80.31 | 0.75 | 613.35 | 14.73 | 3.14 [c] (± 0.51) | 11.42 | 941 |
| Water | 100 | 100 | 1.00 | 15.30 | 0.37 | 0.32 [d] (± 0.11) | 1.16 | 34 |
| Unidentified feature | | | | 40.47 | 0.97 | – | –. | – |
| Total (Area of MZ-1) | | | | 4164 | 100 | 27.49 | 100 | 8562 |

Note: Features assessed with supervised classification and area estimates of features (mean ± standard error) by pixel counts (N = 8562). [†] Overall accuracy of features by supervised classification: 82.0%, Kappa index 0.8. [††] Means with the same letter in the same column are not significantly different by Duncan's Multiple Range Test at $\alpha = 0.05$.

*3.2. Relationship Among Features*

To determine the extent to which features interrelate, the pixel counts of the features were assessed with multivariate analysis. In this case, in order to establish the coexistence of monuments with these features, the number of monuments was taken into account in the analysis. First, a correlation matrix showed that the crop field did not significantly relate to other features (Table 2). On the contrary, the semi-natural woody vegetation strongly and positively correlated with the shrub-prone patches ($p < 0.0001$) and the monuments ($p < 0.05$). The shrub-prone patches again were in positive correlation with the exposed land ($p < 0.05$). The results showed that (1) the crop field is independent of the other features, (2) the semi-natural woody vegetation is dependent on the expansion of the shrub-prone patch, (3) the semi-natural woody vegetation is a significant impact on the monuments, and (4) the shrub-prone patch is triggered by the increment of the exposed land. Second, the principal component analysis dropped the first four components (PC) of Eigenvalues above 1, which gave 83% of the overall variance (Table 3). The PC 1 accounted for 31.2% of variation among the features, and those with 50% and above loading to the PC were considered interpreting the component. The semi-natural woody vegetation and the shrub-prone patches loaded similarly, at approximately 60% each to the PC 1, accordingly explaining the existence of "unmanaged vegetated cover" at the study site. In PC 2, the

loading of the built-up feature and the monument was 55% and 50%, respectively, whereas exposed land contrasted them. The argument stresses the PC 2 as "anthropogenic artefacts", in which expansion of the former two is a trigger to decrease the size of the exposed land. In PC 3, the crop field was the only feature that loaded above 80%, thereby representing "agricultural land". In PC 4, the water indicated the highest loading, at 85%; in consequence, the result suggests the existence of a "reservoir" or "tank". The four PCs clarified the appearance that the "unmanaged vegetated cover" was the most prominent land cover of the contemporary landscape of the MZ-1, while there coexists a certain degree of the "anthropogenic artefacts", a little proportion of the "agricultural land" as well as the "reservoir".

**Table 2.** Pearson's correlation matrix expressed relation of landscape features, including monuments at the Monumental Zone-1.

| Feature | Feature | | | | | | |
|---|---|---|---|---|---|---|---|
| | Crop Field | Semi- Natural Woody Vegetation | Shrub- Prone Patch | Built-Up Feature | Exposed Land | Water | Monument |
| Crop field | 1.00 | 0.23 | –0.15 | 0.04 | –0.05 | –0.09 | –0.19 |
| Semi-natural woody vegetation | 0.23 | 1.00 | 0.67 ** | 0.20 | 0.28 | 0.12 | 0.40 * |
| Shrub-prone patch | –0.15 | 0.67 ** | 1.00 | 0.14 | 0.49 * | –0.07 | 0.32 |
| Built-up feature | 0.04 | 0.20 | 0.14 | 1.00 | –0.15 | –0.06 | 0.31 |
| Exposed land | –0.05 | 0.28 | 0.49 * | –0.15 | 1.00 | –0.01 | –0.12 |
| Water | –0.09 | 0.12 | –0.07 | –0.06 | –0.01 | 1.00 | 0.09 |
| Monument | –0.19 | 0.40 * | 0.32 | 0.31 | –0.12 | 0.09 | 1.00 |

* Significance at $p < 0.05$; ** $p < 0.0001$.

**Table 3.** Principal component (PC) loadings of landscape features at the Monumental Zone-1.

| Feature | PC 1 (31.2%) (Eigenvalue = 2.18) | PC 2 (19.86%) (Eigenvalue = 1.39) | PC 3 (16.89%) (Eigenvalue = 1.18) | PC 4 (14.72%) (Eigenvalue = 1.03) |
|---|---|---|---|---|
| Crop field | –0.04 | –0.04 | 0.83 | 0.36 |
| Semi-natural woody vegetation | 0.58 | –0.01 | 0.24 | 0.26 |
| Shrub-prone patch | 0.59 | –0.20 | –0.06 | –0.17 |
| Built-up feature | 0.23 | 0.55 | 0.23 | –0.22 |
| Exposed land | 0.32 | –0.63 | –0.06 | –0.06 |
| Water | 0.05 | 0.08 | –0.39 | 0.85 |
| Monument | 0.40 | 0.50 | –0.21 | –0.01 |

Note: Overall variance of four principal components: 83%. Variance explanations and Eigenvalues of the PCs are shown in parentheses.

## 4. Discussion

Contemporary landscape structures of the MZ-1 comprise six land covers, referred to as "features" in Table 1. The features reflect human-induced landscapes of the zone, where lands used for agriculture, buildings, roads, and reservoirs coexist with unmanaged vegetated lands.

The uniqueness of the MZ-1 is the presence of ten-century-old monuments, clustered across the zone [11,18], which are viewed in coexistence with other human-induced landscape features. In the vicinity of the monuments are the farmlands, where the local settlers cultivate peanuts, sesame, mung beans, pigeon peas, and sorghum for their major livelihood. The crops sown at the study site are similar to those grown in other parts of the Dry Zone of Myanmar [20,22], and they are the leading agricultural species [21]. Crop fields characterize land use activity by the local farmers during the farming season from May to December each year. The proportion of the land with crop coverage in this study, however, reflects 7.5%, which is apparently half of the exposed (ploughed) land surface area. The results suggest that agricultural land is incompletely covered with crops, while some areas remain uncultivated, even during the cropping season. Field surveys conducted by the authors agree with the data regarding uncultivated acreage at the study site. This is likely due to various socio-economic factors of local land users. It is questionable whether the presence of the uncultivated fields downgrades the landscape of

the MZ-1; nevertheless, it is an issue for future research and analysis. A tourism sector development project implemented in very recent years promotes the unique landscape of Bagan for the coexistence of its agricultural lands with its monuments [34]. Interestingly, the crop fields are independent of the monuments and other existing features, and this reveals that the presence of the cultivated fields in particular has little or no impact on the monuments in terms of land use.

The exposed land that is recognized as a part of the agricultural area, on the other hand, is in a positive relationship with the shrub-prone patches. The land appears to have been left uncropped after ploughing until the end of September, which is the later part of the monsoon season (mid-May to October). It seems that rainfall at the study site did not provide sufficient moisture for crop growing as the crops are exclusively rain-fed, thereby expanding the exposed land surface to about 15% of the MZ-1 (Table 1). Receiving erratic rainfall during the crop season has driven the failure of crop yield in an area of the Dry Zone [22]. This might have led to a socio-economic impact on the livelihood of local farmers. In the case of leaving the land exposed, the incidence of shrubs appears to conquer such land surfaces by means of natural succession. The land, in consequence, would have transformed into shrub-prone area, and the data of the current study suggest that the more the exposed land, the larger the expansion of the shrub-prone patch.

The second largest area of shrub-prone patch at the study site implies that the massive proportion of cultivatable lands has been recently left with no more crop cover after ploughing or previous sowing. Under favourable conditions, those lands are subject to the regeneration of shrubs, which gradually invade the entire exposed land surface. Within a short duration of exposure after crop activity, the degree of shrub incidence on the land is found to be high [35]. If continuous cultivation is practiced in each growing season, ploughing the earth's crust is a means of destroying shrub propagules and would reduce the shrub-invaded lands. The shrub-invaded patches may have functioned as storehouses for disseminated propagules of woody plant species in the surrounding areas. This argument is supported by reports on increasing plant species and density, including woody plants, due to the presence of shrubs in a particular habitat [36,37]. The shrub-prone patch supports growth of woody plants that, unless encountered by further human intervention, would transform into semi-natural woody vegetation. The results of the current study also suggest that the shrub-prone land is a trigger for the incremental spread of the semi-natural woody vegetation. The data agreed with the observations of Wang et al. (2016) [38] that a forest regenerates by natural means of grass and shrub covers via land use change after agricultural activity.

The stretch of the semi-natural woody vegetation at the MZ-1 is astonishingly large, covering 37% of the zone (Table 1). Since it is a time-course factor, massive expansion of such vegetation in the contemporary landscape shows that little or no management has been applied to the shrub-prone patches developed from the exposed lands in recent years. Local xeromorphic woody plants such as *Acacia catechu*, *Acacia leucophloea*, *Borassus flabellifer*, and *Zizyphus jujuba* are common to the study site, whereas *Prosopis juliflora* has become an invasive species widely across the Dry Zone of Myanmar [39,40]. The *P. juliflora* is a robust, semi-arid tree species that has an outstanding characteristic of invasion into natural and/or human-induced landscapes [41]. However, no data of its location-specific invasion to the landscape of the MZ-1 are available for the current study. A remarkable result of the relationship between the semi-natural woody vegetation and the monument, instead, reveals that the area's expansion of woody plant species has been extending to the environs of the monuments. If the plants grow close to the monuments, the root extension could penetrate the structures, inducing flaws and cracks. Under severe conditions, a collapse of the monuments might lead to the loss of the historical symbols.

The existence of the built-up features and the water is also independent of others at the MZ-1. The built-up features are mainly seen as settlement areas, paved roads, and monumental compounds occupying 13% of the zone, while the waters as the reservoirs are the smallest occupancy, at less than 1%. The data suggest that the MZ-1 is not yet very urbanized, although there are anthropogenic features integrating into the landscape.

This study was unable to interpret the monuments, both in the 30-m and the 15-m resolution images of Landsat 8, probably due to insufficient image quality to access the monuments. Field-recorded GPS points of monument locations and/or those accessed in a 2-m resolution Google Earth map (CNES/ Airbus, 2018), however, accounted 2348 monuments scattered within the MZ-1. This revealed that ground-truth data are of the utmost importance to validate the monuments in this study. Of the geolocated monument points plotted on the pan-sharpened image, they superimposed the built-up features and the semi-natural woody vegetation. Crosscheck in Google Earth clarified that some monument points in the range of 1–6 were in the enclosures designated as monumental compounds, where platforms and developed structures stretched in proximity to the monuments. The enclosures spotted in and outside the residential areas, and some intermingled with the semi-natural vegetation. The locations of the monument enclosures were also checked in the inventory maps of Pichard (1992–1995) [12]. Validations supported interpreting the monumental compounds as the built-up feature.

The close relationship existing between the shrub-prone patches and the semi-natural woody vegetation accounts for the prominence of "unmanaged vegetated cover" at the study site. The argument illustrates on the one hand the tendency for land to transform into a natural state; on the other hand, it points to the amendment of the current landscape as a safeguarding measure for protection of the historical setting. Reflecting poor site management apparently, the expansion of unmanaged vegetation within the MZ-1 would become an interference for access to the monuments. The threat may also downgrade the attractiveness and magnificence of the cultural landscape. Stands of tall woody plants are supposedly an obstruction to the view and to the monuments. Erect above the small-sized monuments, tree-type vegetation would hinder the structures and seriously damage the monuments by root penetration. Propagules disseminated by such vegetation to the surrounding area are an important source of regeneration over the landscape. Cultivation activities on the farmland are creditable to suppress the sprouts and seeds of those plants, mainly in the rainy season. Successive controls in every onset of rain by agricultural land management of local farmers are highly appreciated as a matter of timely site management. Since the MZ-1 is of a historically developed landscape, anthropogenic modifications over time have remade the landscape as part of the cultural identity. In short, coexisting with the built-up features, referred to as "anthropogenic artefacts", "agricultural land", and "reservoir", proper management of semi-natural vegetation, would improve the value of the landscape. Through enhancing living landscape, local communities, pilgrims and visitors, and, collectively, the global society are expected to receive ecosystem services directly or indirectly from the heritage site in the Dry Zone of Myanmar.

## 5. Conclusions

The contemporary landscape of the MZ-1 is predominantly structured with (1) vegetated land covers or features of crop field, semi-natural woody vegetation, and shrub-prone patches; and (2) non-vegetated built-up features, exposed land, and water. The land use for the crop field quantifies a small proportion, compared to the 4–6 times larger coverage of the other two vegetated features. The built-up feature and the exposed land also occupy the amount of area similar to that of the crop field, whilst the water makes up the smallest fraction. In land-use aspects, crop cultivation on agricultural land is found to have depreciated, while the exposed land surface remains as large as the crop-covered area. Low occupancy of the built-up structures reflects the fact that the zone is not highly populated, perhaps still in the (traditional) rural landscape. The contrasting areas of features yield to the evaluation of their coexisting relationships in the MZ-1.

Independence of the existing crop field indicates that agricultural area has no impact on other features, particularly on monuments at the MZ-1. However, the semi-natural woody vegetation develops in the environs of the monuments and seems to be a threat to them. The growth of this vegetation is triggered by the occurrence of the shrub-prone patch; the latter, in turn, is driven by the expansion of the exposed lands. In this regard, agricultural land use and management and the

coverage of semi-natural vegetation relative to the monument areas in such landscapes are significant matters for further investigation. The remarkable prominence of "unmanaged vegetated cover" coexisting with "anthropogenic artefacts", "agricultural land", and "reservoirs" draws attention to timely amendments of the human-induced landscape as safeguarding measures for the historical settings in Bagan. Provided that no accessibility to the locations of the monuments was available, the assessment of the relationship between the monuments and the other features will not be given any account in the current study.

The study deduces management-related features of the contemporary landscape, stressing priority to locally best-fit strategies for long-term management of tangible and intangible cultural heritage, through broadly covered national law. The study's results emphasize the role of agricultural land that primarily matters for safeguarding the heritage site. Coverage of cultivated crops over the farmlands in corresponding growing seasons will be a breakthrough of strengthening the living landscape in the dry zone ecosystem. Suggesting governmental policy implication and supports, the study urges the intrinsic involvement of local stakeholders in the holistic management of the entities and the site.

**Author Contributions:** M.Z.N.A. wrote the manuscript, and S.S. supervised, reviewed, and edited it.

**Funding:** This research was funded by the Japan International Cooperation Agency (JICA), titled the Long-term Training Programme for Academic Staff in Yezin Agricultural University for Postgraduate Degree (Ph.D.) in Japan. And the APC was funded by the Kyoto University.

**Acknowledgments:** The authors gratefully acknowledge miscellaneous support from the Technical Cooperation Project of Yezin Agricultural University and Japan International Cooperation Agency (YAU-JICA TCP) while travelling to the research site. Contributions and assistance of officials as well as staff members of the Department of Agriculture (Ministry of Agriculture, Livestock and Irrigation) and the Department of Archaeology and National Museum (Ministry of Religious Affairs and Culture), the Republic of The Union of Myanmar, are also appreciated for acquiring ground-truth data, which are invaluable sources for analyses.

**Conflicts of Interest:** The authors declare no conflict of interest.

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
