# Peer review of "Contemporary Landscape Structure within Monumental Zone-1 at Bagan Cultural Heritage Site, Myanmar"

_heritage, doi:10.3390/heritage2020107_

Round 1
Reviewer 1 Report
I think it is a very valuable research. It presents opportunities for a long-term analysis of environment changes in the heritage area with the prospects for management strategy proposals.
Author Response
Response to Reviewer 1 Comments
Point 1: I think it is a very valuable research. It presents opportunities for a long-term analysis of environment changes in the heritage area with the prospects for management strategy proposals.
Response 1: The reviewer’s comment motivated the authors and is greatly acknowledged. The research was, of course, intended to address holistic management strategies for the cultural heritage site, integrating all possible means through scientific and policy guidelines. Not only the tangible, but also the intangible heritage such as traditional agricultural practices and local social-economic activities, need be intermingled for safeguarding the site in landscape scale. Cultivations in the environs of the monuments reflect a key driver for impressive living landscape of the site, and thus, long-term research-based monitoring will provide environmentally friendly strategies for safeguarding the heritage as well as the site. The analytical outcomes will shed concrete proposals and guidelines of the site management, to stakeholders, mainly to the local communities.
Reviewer 2 Report
After reviewing the interesting paper entitled “Contemporary Landscape Structure within Monumental Zone-1 at Bagan Culture Heritage Site Myanmar”, it raises some questions that the authors should address at the beginning (objectives) and as results (conclusions) regarding the Study.
How this pure analytical study is related to any attempt to improve the real management of the Site?
What is the intention or original idea lying behind the study?
How this study may affect standards or rules that regulate the protection of the Site?
In the end, how is this study useful (can be of application) for the real situation of the Site?
Author Response
Response to Reviewer 2 Comments
Point 1: After reviewing the interesting paper entitled “Contemporary Landscape Structure within Monumental Zone-1 at Bagan Culture Heritage Site Myanmar”, it raises some questions that the authors should address at the beginning (objectives) and as results (conclusions) regarding the Study.
Response 1: The reviewer’s comments and suggestions are valuable to improve the manuscript. The following will be integrated to the introductory part (objectives) and conclusions of the manuscript to provide the readers more understandable information.
Point 2. How this pure analytical study is related to any attempt to improve the real management of the Site?
Response 2: The study integrates tangible and intangible cultural heritage by the important role of site management. In reality, conservation measures for the monuments are well implemented by the state government and some local non-governmental associations. Rather, the environmental setting of the monuments, to a great extent, need prioritize to safeguard all the cultural heritage. Massive expansion of unmanaged vegetated cover across the MZ-1, through the current study, apparently elevates the threats to the site, exclusively to the centuries-old monuments. Any promoting effort to fully cover the agricultural land by seasonal cultivations is a measure to suppress the weedy and shrub influences, destroying their propagules. The study convincingly proposes the growth of the vegetation to be under management to sustain the integrity of the historical landscape.
Point 3: What is the intention or original idea lying behind the study?
Response 3: The underlying idea of the study is to promote agricultural land management as a complement to tangible as well as intangible heritage through ecological approach in landscape scale in the Bagan Cultural Heritage Site.
Point 4: How this study may affect standards or rules that regulate the protection of the Site?
Response 4: The analytical results of the study deduce management-related characters in the landscape scale of the MZ-1. These characters in the contemporary landscape constructively point out fitting locally adapted explicit strategies for managing respective elements within the zone although the national law broadly covers conservation and preservation of the cultural heritage and the site.
Point 5: In the end, how is this study useful (can be of application) for the real situation of the Site?
Response 5: 1) The results of the study are also feasible to considering current and long-term site management plans, highlighting the role of agricultural land that primarily matters for conserving the heritage site. 2) Suggesting governmental policy implication and supports, the study urges intrinsic involvement of local stakeholders in collaborative approach to conserve the site and the entities under holistic management.
Reviewer 3 Report
A novel article using satellite imagery to examine land use at Bagan CHS, which draws some interesting conclusions for land practice and management. It’s not entirely clear what the study goal is until the conclusion, but this is a minor structural issue. It takes a new approach, and is well written, clearly structured, and the conclusions are largely consistent with the evidence and arguments presented, although the importance of those conclusions should be emphasised more strongly.
There are several minor points that could be clarified to strengthen the article.
1) Structural points:
a. The main research question could be more clearly addressed early on. It is clear what the authors are doing, but not why, or why it matters: this does not really become clear until the conclusion. A clearer focus on the relationship between land management and heritage management throughout would significantly strengthen the article.
b. There are some very interesting points about the implications for heritage management in the conclusion, but these are rather lost in the technical information about identified land classes. It would be good to emphasise them more.
2) There are several minor methodological areas for improvement
a. Line 103 – what was the resolution of the Google Earth image, and whose was it?
b. Line 117 – the pan-sharpened image: presumably this was sharpened using a panchromatic 15m image from Landsat 8? So the reference image has 15m resolution? This should be clarified.
c. Line 141-143 – The accuracy check was done using the (presumed) 15m resolution pan-sharpened image. However, ref 11 (Hudson) notes that of the 3822 monuments spread across the area (not just MZ1), 2372 are classed as “small”, i.e. below 12m. It would be good to have a greater discussion of the possible impact on the study of the fact that a significant number of the monuments may be too small to be picked up on Landsat. In line 351/2 the authors refer to “monumental compounds” – how do these detect on landsat – are they common? How many of the monuments do they account for?
How may this resolution issue affect the conclusions drawn when a significant number of the monuments may not be visible? For example the authors note:
line 375 “Independence of the existing crop field indicates that agricultural area has no impact on other features, particularly on monuments at the MZ-1.” What if the monuments are not visible?
d. Line 147 – can you provide more information on the accuracy check?
e. Line 358/9 – I’m unclear how the prominence of unmanaged vegetation points to “the amendment of the current landscape as a safeguarding measure for protection of the historical setting”. (repeated in the last line of the conclusion, where it is referred to as a timely amendment)
Would this have been done by the local people? By UNESCO? By the site managers? How does it safeguard it? Why is it timely? Was it deliberate?
f. Line 363 - …”would improve the value of the landscape”. Who for? How?
Author Response
Response to Reviewer 3 Comments
Point 1(a): The main research question could be more clearly addressed early on. It is clear what the authors are doing, but not why, or why it matters: this does not really become clear until the conclusion. A clearer focus on the relationship between land management and heritage management throughout would significantly strengthen the article.
Response 1 (a): The reviewer’s comments were very constructive to maximize the manuscript quality, and highly acknowledged.
The current study was indeed a preliminary step to proceed long-term examinations for provision of holistic management strategies of tangible as well intangible cultural heritage in Bagan. It, therefore, essentially requires integration of all possible means through scientific and policy guidelines. The national government of Myanmar has already implemented law for conservation and preservation of cultural heritage and the site based on UNESCO guidelines Rather it still opens question of site management stressing that environmental setting of the heritage site has any impact on safeguarding the tangible structures. Agricultural lands in the environs of the monuments endow the living landscape of Bagan (ref. 34), and thus management of the agricultural lands are supposed to be a trigger of any influence on the cultural landscape. Long-term research-based monitoring will shed analytical outcomes to formulate locally adapted guidelines and proposals for site management, whereas data to date are prerequisite for succeeding examinations.
Point 1 (b): There are some very interesting points about the implications for heritage management in the conclusion, but these are rather lost in the technical information about identified land classes. It would be good to emphasise them more.
Response 1 (b): Many thanks to the reviewer for suggestions, and the improvement will be fixed in the manuscript.
Point 2(a): Line 103 – what was the resolution of the Google Earth image, and whose was it?
Response 2 (a): The Google Earth satellite image was of 2-m resolution provided by CNES/Airbus (2018).
Point 2(b): Line 117 – the pan-sharpened image: presumably this was sharpened using a panchromatic 15m image from Landsat 8? So the reference image has 15m resolution? This should be clarified.
Response 2 (b): The 15-m resolution panchromatic image of Landsat 8 was used in pan-sharpening and as reference image for accuracy check. It will be calrified in the text.
Point 2(c): Line 141-143 – The accuracy check was done using the (presumed) 15m resolution pan-sharpened image. However, ref 11 (Hudson) notes that of the 3822 monuments spread across the area (not just MZ1), 2372 are classed as “small”, i.e. below 12m. It would be good to have a greater discussion of the possible impact on the study of the fact that a significant number of the monuments may be too small to be picked up on Landsat. In line 351/2 the authors refer to “monumental compounds” – how do these detect on landsat – are they common? How many of the monuments do they account for?
How may this resolution issue affect the conclusions drawn when a significant number of the monuments may not be visible? For example the authors note:line 375 “Independence of the existing crop field indicates that agricultural area has no impact on other features, particularly on monuments at the MZ-1.” What if the monuments are not visible?
Response 2 (c): Line 141–143: The monument counts in the study were of course not through the Landsat 8 image. As reviewer pointed, the Landsat 8 image was not rather supportive to access the monuments. Without ground-truth GPS points and/ or those accessed in 2-m resolution Google Earth Map, no geolocation of monuments can be processed in GIS. The noise observed through 15-m pan-sharpened image was found that the monument points overlapped the land cover classes: semi-natural woody vegetation and built-up features in the study. The number of monuments 2348 in this study represented both of erecting structures and mounds of damaged ones within the MZ-1. The numbers in the references were those not only in the MZ-1, but also in other MZs.
Line 351/352: “monumental compounds” are the signatures generated by semi-automatic classification, which were classed as built-up features. The signatures displayed areas of one to some 30 continuous pixels, which in real land cover represented the concrete platform or other constructed structures within an enclosure. Geolocated monument points overlaid the signatures, and out of the 2348 monument points (Figure 1), few numbers in the range of 1–6, were in the continuous pixel areas. Some were in the monastic enclosures, whereas many scattered in the residential areas. In this regard, the pixel areas of the signatures with monument points overlaid were referred to as “monumental compound” in this study. Field survey and Google Earth Map were helpful in considering the surrounding of the monument points. It was found during field surveys that not every monument was in the enclosure, mostly small ones. The status of monument enclosures was also checked in the inventory maps of Pichard (1992–1995) (ref. 12). Nevertheless, fortifications of many compounds collapsed and were remnants today. In honesty, nor the monuments in such condition were counted, and any data of the number can be shown by the feature “monumental compounds” in the current study, and it is admitted that further examinations are required based on this preliminary study.
Unless point data of the monuments were not processed, there will be no any other option to judge the ‘monuments’ through Landsat 8 image for insufficient resolution to see the monuments in scatter. Therefore, ground-truth data and/ or access via Google Earth Map are of great importance in such assessment. The geolocated points in GIS showed several hundreds superimposed the semi-natural woody vegetation, while little or no points overlaid the crop field. Through pixel counts of the features, tremendous proportion of the pixels representing the semi-natural woody vegetation was in close relation with the monuments, whereas data of crop field did not depict any relation with the monuments. Referring to the statement in line 375, none of the results of multivariate analysis would be interpreted in any relationship with the monuments if the monuments were inaccessible.
Point 2(d): Line 147 – can you provide more information on the accuracy check?
Response 2 (d): Line 147: The methodological approach provided a cross-reference for accuracy of the features by means of manual counting the pixels. Pixels of the features were first marked with points of 50-m distance so that two adjacent pixels of 30-m resolution map can be taken into account. For precise counting, grids were applied to the map, and the size of each grid cell was 1 km2. Within each grid, the pixels of respective features touched with the marked points were counted, and 8562 pixel-counts were accessed in 30 grids, i.e., the grids that partially overlaid the map at the boundary were not considered. The pixels of each feature were, then, computed for area, multiplying the pixel number with 30 x 30 m2 of each pixel size. The pixel number and area measures also validated accuracy of each feature classified by supervised classification.
Point 2(e): Line 358/9 – I’m unclear how the prominence of unmanaged vegetation points to “the amendment of the current landscape as a safeguarding measure for protection of the historical setting”. (repeated in the last line of the conclusion, where it is referred to as a timely amendment)
Would this have been done by the local people? By UNESCO? By the site managers? How does it safeguard it? Why is it timely? Was it deliberate?
Response 2 (e): Efforts of managing the unmanaged vegetation currently are seemingly not well implemented in the MZ-1 because the results of land cover classification and pixel counts indicated immense acreage of such feature. The growth of weedy and shrub-like vegetation across the land of the MZ-1 would become an interference for access, particularly of pilgrims and visitors, to the monuments, and they may degrade the attractiveness and splendor of the cultural landscape. This is because many access-ways to the monuments pass through the farmlands which were observed in field, and can even be checked in Google Earth Map. The woody or shrub-prone patch may host the propagules of other shrub or woody plants, which will later develop into trees; other authors also pointed such condition (ref. 36, 37, 38).
The stands of the woody plants would obsrtuct the view and the monuments. The height of tree-type vegetation will possibly hinder small-sized monuments, and seriously their root growth penetrating into the structures, regardless of the size, is a harm to the integrity of the centuries-old monuments. The unmanaged vegetation cover is living and disseminating their propagules to various directions. Such vegetation need be controlled to limit the growth and multiplication corresponding to their seasonal development. Cultivation activities on the farmland is creditable to suppress sprouts of the undesirable plants, mainly in the rainy season when crop cultivations are normally practiced in the study area. Reducing the host habitat for seeds dispersed from shrubs or trees nearby will gradually limit the stretch of the semi-natural woody groves. Controls in every onset of rain in successive years are recommended, and thus, such measures are referred to as “timely amendment” of the unmanaged vegetation. Unless otherwise, they will degrade the integrity of the MZ-1 landscape which had been culturally developed by anthropogenic activities through years. Amendment of current unmanaged vegetation and timely management activities in the long run will safeguard the heritage as well as the cultural landscape.
In the report titled with “Archaeological Conservation of Bagan Ancient Monuments in Myanmar” (ref. 10), the author also described the threats of the vegetation to the monuments and the conservation practices (ref. 10, pp. B29–B30). Conserving the monuments and the site is chiefly administered by the Department of Archaeology and National Museum based on the UNESCO guidelines, when necessary, other authorities concerned, and local communities participate in management activities.
The current study suggests collaborative approach of stakeholders, mainly local communities, in managing the site for safeguarding tangible as well as intangible cultural heritage. In this context, agricultural land management of local farmers are highly appreciated as they are those who invest daily work powers and hours in the farms in the surrounding of the monuments. As mentioned above, seasonal land management by agricultural activities of local people is an endeavour to overcome the noise of the unmanaged vegetation.
Point 2(f): Line 363 - …”would improve the value of the landscape”. Who for? How?
Response 2 (f): Feasibility of the current study is support of analytical results which can integrate into considering current and long-term management plans of the cultural heritage site. Stressing the role of agricultural land, strengthening agricultural activities of local land users, for full coverage of crops over the farmlands in corresponding cropping seasons, is the most creditable means for enhancing the living landscape in the dry-zone ecosystem. Safeguarding tangible entities in one hand, enhancing intangible traditional practices in the other, will promote the values of the ecosystem with premium services. The local communities are exclusively those who in majority rely on such environment for daily lives. Pilgrims and visitors, who enjoy the scenic landscape and the architectural structures, perhaps receive foods and other services during the stay. Collectively, the global society directly or indirectly relates with the cultural landscape of many monuments and other human-induced artefacts in the semi-arid ecosystem of Myanmar in the mainland South-east Asia.
Reviewer 4 Report
This is an interesting study, based on a holistic approach. It well describes cultural premises, methodologies, scientific tools, and the results are convincingly presented and explained.
The paper is well written the English is good.
There are a few minor revision that should be made for it to be accettable for publishing.
I'd like to suggest a better development of the conclusions. This section needs to be integrated by giving more suggestions about the possible development of strategies for the enhancement of those sites characterized by special environmental settings and cultural heritage.
Author Response
Response to Reviewer 4 Comments
Point 1: This is an interesting study, based on a holistic approach. It well describes cultural premises, methodologies, scientific tools, and the results are convincingly presented and explained. The paper is well written the English is good. There are a few minor revisions that should be made for it to be acceptable for publishing.
I'd like to suggest a better development of the conclusions. This section needs to be integrated by giving more suggestions about the possible development of strategies for the enhancement of those sites characterized by special environmental settings and cultural heritage.
Response 1: It is very grateful and helpful suggestions the reviewer supported. The following will be integrated to the conclusions of the manuscript.
The analytical results of the study deduce management-related characters in the landscape scale of the MZ-1. These characters in the contemporary landscape constructively point out fitting locally adapted explicit strategies for managing respective elements within the zone although the national law broadly covers conservation and preservation of the cultural heritage and the site. The results of the study are also feasible to considering current and long-term site management plans, highlighting the role of agricultural land that primarily matters for conserving the heritage site. Strengthening agricultural activities for full coverage of the crop fields in corresponding cropping seasons is the most creditable means of maintaining the living landscape in the dry-zone ecosystem. Safeguarding tangible entities in one hand, enhancing intangible traditional practices, in the other, will promote the values of the ecosystem with premium services. Suggesting governmental policy implication and supports, the study urges intrinsic involvement of local stakeholders in collaborative approach of conserving the site and the entities under holistic management.